# External validation of postnatal gestational age estimation using newborn metabolic profiles in Matlab, Bangladesh

Malia SQ Murphy[1], Steven Hawken[1,2], Wei Cheng[1], Lindsay A Wilson[1], Monica Lamoureux[3], Matthew Henderson[3], Jesmin Pervin[4], Azad Chowdhury[5], Courtney Gravett[6], Eve Lackritz[6], Beth K Potter[2], Mark Walker[1], Julian Little[2], Anisur Rahman[4], Pranesh Chakraborty[3], Kumanan Wilson[1,2]*

[1]Clinical Epidemiology Program, Ottawa Hospital Research Institute, Ottawa, Canada; [2]Department of Epidemiology and Community Health, University of Ottawa, Ottawa, Canada; [3]Newborn Screening Ontario, Children's Hospital of Eastern Ontario, Ottawa, Canada; [4]International Centre for Diarrhoeal Disease Research, Dhaka, Bangladesh; [5]Dhaka Shishu (Children) Hospital, Dhaka, Bangladesh; [6]Global Alliance to Prevent Prematurity and Stillbirth, Lynnwood, United Stares

*For correspondence:
kwilson@ohri.ca

Competing interests: The authors declare that no competing interests exist.

**Abstract** This study sought to evaluate the performance of metabolic gestational age estimation models developed in Ontario, Canada in infants born in Bangladesh. Cord and heel prick blood spots were collected in Bangladesh and analyzed at a newborn screening facility in Ottawa, Canada. Algorithm-derived estimates of gestational age and preterm birth were compared to ultrasound-validated estimates. 1036 cord blood and 487 heel prick samples were collected from 1069 unique newborns. The majority of samples (93.2% of heel prick and 89.9% of cord blood) were collected from term infants. When applied to heel prick data, algorithms correctly estimated gestational age to within an average deviation of 1 week overall (root mean square error = 1.07 weeks). Metabolic gestational age estimation provides accurate population-level estimates of gestational age in this data set. Models were effective on data obtained from both heel prick and cord blood, the latter being a more feasible option in low-resource settings.
DOI: https://doi.org/10.7554/eLife.42627.001

## Introduction

Complications related to preterm birth are the leading cause of death among children under 5 years of age (*March of Dimes, 2012*). Estimating the burden of preterm birth in low-resource settings is challenging due to the absence of ultrasound technology and the unreliability of recall of last menstrual period. Commonly used estimates obtained late in gestation or postnatally (e.g. fundal height, and Ballard or Dubowitz scores) are subject to high inter-user variability and poor reliability in small for gestational age and preterm infants (*Taylor et al., 2010*; *Spinnato et al., 1984*; *Robillard et al., 1992*). In addition, data on preterm birth are not routinely documented in some countries and may not be classified according to international standards (*Quinn et al., 2016*), thus impeding the development of strategies for resource allocation to support global and local health initiatives. Strengthened data surveillance systems to more accurately assess and track changes in preterm birth across jurisdictions are urgently required (*March of Dimes, 2012*).

Algorithms based on newborn metabolic profiles in combination with clinical covariates such as sex and birthweight have demonstrated the potential to accurately categorize infants across preterm birth categories in high-resource settings (*Jelliffe-Pawlowski et al., 2016*; *Ryckman et al., 2016*;

**eLife digest** Complications from preterm birth are the leading cause of death among children under five. Ultrasounds are routinely used in wealthy countries to track babies' development. In countries with limited resources, however, ultrasounds are rare, making it harder to estimate how many children are born prematurely. Blood tests may offer a way to determine whether a newborn was born too early when ultrasounds are not available. Many countries already require clinicians to collect a drop of blood from newborns via a heel-prick or from their umbilical cord. Testing these blood spots identifies babies at risk of rare conditions so they can receive prompt treatment.

Chemicals in the blood vary depending on how long the newborn spent growing in its mother's womb. Scientists have developed a mathematical formula that can estimate a baby's gestational age based on these chemicals. Using blood spots to estimate gestational age worked well when this strategy was tested in Canada, a high-income country. More tests are needed to determine if it works in low-income countries.

Now, Murphy et al. show their blood spot-testing strategy also reliably predicts the gestational age of babies in Matlab, Bangladesh. In the experiments, blood spots were collected from 1,069 newborns. This included 1,036 cord blood samples and 487 heel prick samples. Nearly all the samples came from full-term infants. A mathematical model estimated the infants' gestational age to within an average of one week of their true age when applied to heel-prick blood samples and to within two weeks of the baby's true gestational age 94% of the time.

The model also provided reliable estimates of babies' gestational ages when cord blood samples were tested, which is useful as the Bangladeshi parents were more comfortable with this method of blood collection. Using this strategy to estimate how many babies are born too early in low-income countries may help the countries develop strategies to reduce preterm births. The estimates might also help identify preterm babies who need special care.

DOI: https://doi.org/10.7554/eLife.42627.002

*Wilson et al., 2016*). Data from newborn screening programs in North America have been used to create models capable of estimating gestational age to within 1–2 weeks, but their performance among other infant populations is uncertain. Recent work has focused on refining these models and tailoring them for use across a range of environments and sub-populations, and has suggested that while the models perform well among infants from a variety of backgrounds, ethnicity-specific models may improve the models' performance (*Wilson et al., 2017*; *Hawken et al., 2017*). More recent model iterations have been strenthened by the addition of variables such as newborn hemoglobin peak percentages (calculated from the ratio of fetal to adult hemoglobin levels), which have demonstrated strong associations with gestational age (*Wilson et al., 2017*). While these algorithms have the potential to provide reliable population estimates of preterm birth burden where prenatal ultrasound data are not available, the models' generalizability to all infant populations, as well as the feasibility of collecting samples for analysis in low-resource settings, is uncertain. In this paper, we explore the performance of gestational age estimation models in an infant population born in Matlab, Bangladesh. We also comment on the effect of timing of sample collection on newborn metabolic profiles and the feasibility of newborn blood sample collection and analysis in this setting.

## Patient characteristics

One cord blood sample was excluded because 100% of analyte values were missing. Imputation was conducted for the remaining samples missing analyte values (n = 28 heel samples and 21 cord samples; no individual sample had more than 5/47 (11%) analyte values missing). The final cohort consisted of 1523 samples from 1069 unique individual newborns. 1036 samples were collected immediately after birth (range: 0 min - 2 hr 1 min) from the umbilical cord, and 487 heel prick samples were collected an average of 14 hr 58 min after birth (range: 25 min - 40 hr 30 min). The majority of samples received (93.2% of heel prick samples; 89.9% of cord blood samples) were from term infants (gestational age ≥37 weeks). 18.1% of heel prick samples and 15.9% of cord blood samples were derived from infants with a birthweight <2500 g. Of the 1069 infants included in the study, 454

contributed both heel and cord blood samples. A summary of participant demographics is provided in *Table 1*.

## Performance of gestational age estimation models using heel prick data

We determined the performance of previously published metabolic gestational dating algorithms in heel prick-derived data from the Bangladeshi infant cohort. Results of linear regression analyses for heel prick metabolic profiles demonstrated optimal performance among term infants between 38 and 39 completed gestational weeks (*Figure 1*). Residual plots for each of the three models in both heel and cord samples are provided in *Figure 2*. In general, all models predicted gestational ages close to full term with the highest accuracy, while tending to overestimate gestational age in preterm infants and underestimate gestational age in post-term infants, in the Bangladesh cohort.

A baseline model including only clinical covariates (infant sex, birthweight and multiple birth status, Model 1) provided the least accurate estimation of gestational age relative to ultrasound-validated gestational age estimates, RMSE 1.46 weeks. By comparison, a model including analyte

**Table 1.** Characteristics of infants and samples obtained from them.

| | Heel samples (n = 487) | Cord samples (n = 1036) | Paired heel and cord samples (n = 454 pairs) |
|---|---|---|---|
| Completeness of analyte data[†], n (%) | | | |
| No missing analytes | 459 (94.3%) | 1015 (98.0%) | 427 (94.1%) |
| ≥1 analyte missing, missing values imputed | 28 (5.7%) | 21 (2.0%) | 27 (5.9%) |
| Sex, n (%) | | | |
| Male | 246 (50.5%) | 538 (51.9%) | 234 (51.5%) |
| Female | 241 (49.5%) | 498 (48.1%) | 220 (48.5%) |
| Gestational Age (wks), overall mean (SD) | 39.1 ± 1.5 | 39.0 ± 1.7 | 39.2 ± 1.4 |
| Gestational Age Category (wks$^{days}$), n (%) | | | |
| ≥37 weeks | 454 (93.2%) | 931 (89.9%) | 425 (93.6%) |
| $32^0$-$36^6$ weeks | 32 (6.6%) | 102 (9.8%) | 29 (6.4%) |
| <$32^0$ weeks | 1 (0.2%) | 3 (0.3%) | 0 (0.0%) |
| Birth Weight (g), mean (SD) | | | |
| Overall | 2837.8 ± 433.7 | 2862.1 ± 445.9 | 2846.8 ± 414.0 |
| Term infants only | 2879.5 ± 392.9 | 2916.5 ± 401.7 | 2879.2 ± 389.9 |
| Preterm infants only | 2264.2 ± 554.8 | 2380.3 ± 524.5 | 2372.1 ± 470.4 |
| Birth Weight Category, n (%) | | | |
| ≥4000 g | 3 (0.6%) | 15 (1.5%) | 3 (0.7%) |
| 2500 g to < 4000 g | 396 (81.3%) | 856 (82.6%) | 374 (82.4%) |
| 1500 g to < 2500 g | 84 (17.3%) | 158 (15.2%) | 75 (16.5%) |
| 1000 g to < 1500 g | 4 (0.8%) | 4 (0.4%) | 2 (0.4%) |
| <1000 g | 0 (0.0%) | 3 (0.3%) | 0 (0.0%) |
| Multiple Birth, n (%) | 7 (1.4%) | 19 (1.8%) | 8 (1.8%) |
| Newborn age at sample collection (hrs), mean (SD) | | | |
| Overall | 14.97 ± 6.54 | 0.06 ± 0.25 | 15.06 ± 6.38 (heel) 0.06 ± 0.25 (cord) |
| Term infants only | 14.74 ± 6.42 | 0.06 ± 0.25 | 14.86 ± 6.22 (heel) 0.06 ± 0.25 (cord) |
| Preterm infants only | 18.00 ± 7.50 | 0.09 ± 0.28 | 17.97 ± 7.93 (heel) 0.07 ± 0.26 (cord) |

Data are presented as mean±standard deviation unless otherwise specified. [†]One cord blood sample was excluded in the data preparation step because 100% of analyte data was missing). All other samples with missing analyte data had no more than 5/47 (11%) missing analyte predictors.

DOI: https://doi.org/10.7554/eLife.42627.003

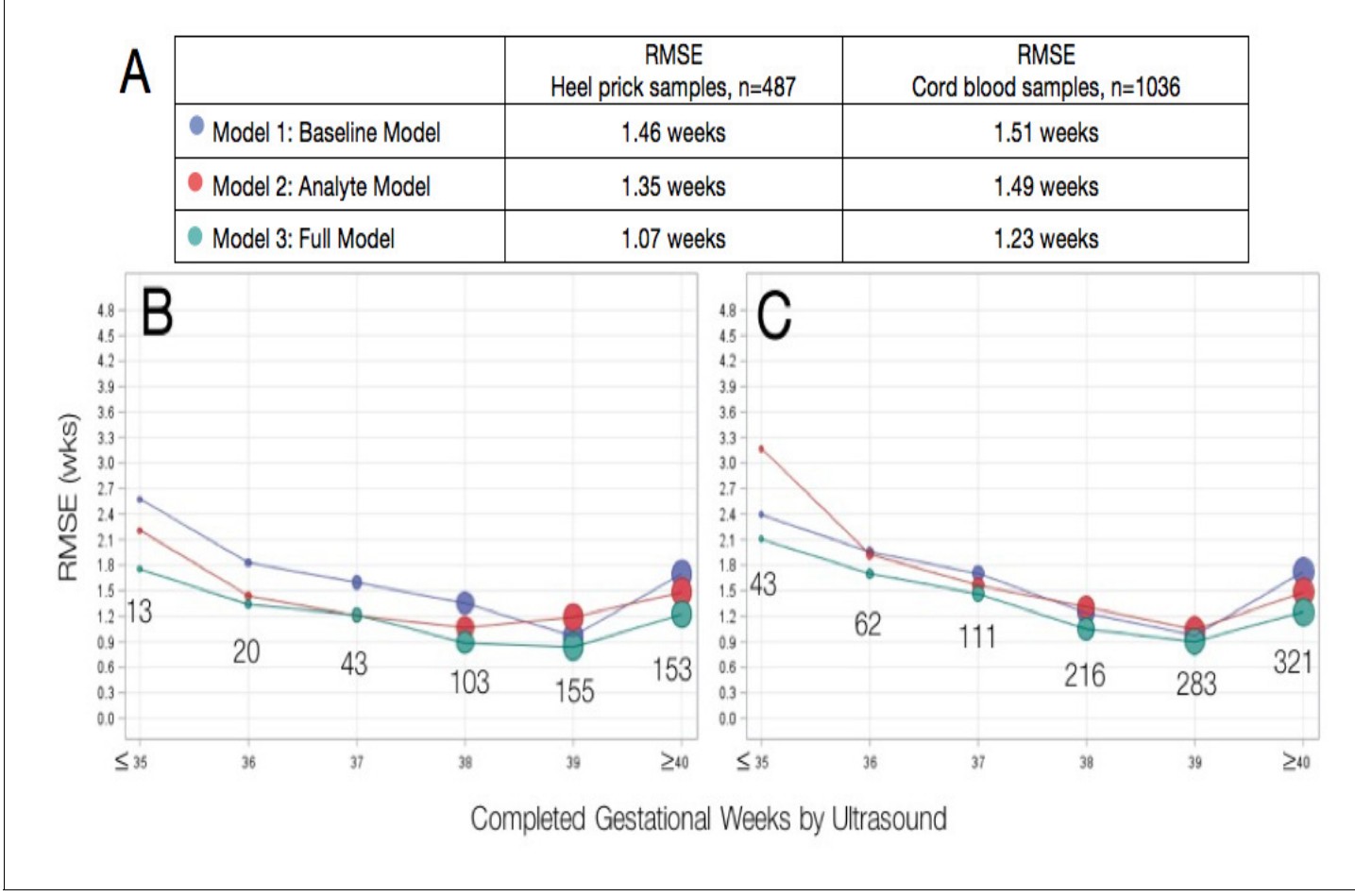

**Figure 1.** Agreement between algorithmic estimates of gestational age compared to ultrasound-validated gestational age. (A) Comparison of overall RMSE for heel prick sample and cord blood samples across gestational age models. Performance of gestational age models by infant birthweight for (B) heel prick samples and (C) cord blood samples. Sample sizes are denoted in the graphs. RMSE, root mean square error (average absolute deviation of observed vs. predicted gestational age in weeks). Reported results are the average over 10 imputations.
DOI: https://doi.org/10.7554/eLife.42627.004

covariates (Model 2) had an RMSE of 1.35 weeks. A full model containing all clinical and analyte data (Model 3) demonstrated the lowest RMSE (best performance) of 1.07 weeks and correctly estimated gestational age to within 1 week for 63.9%, and within 2 weeks for 94.3% of all heel prick samples. Among small for gestational age infants, the full heel prick model had an RMSE of 1.12 weeks when growth restriction was defined as birthweight below the 10th percentile for gestational age and an RMSE of 1.30 weeks when defined as birthweight below the 3rd percentile for gestational age. By these definitions, our model accurately estimated gestational age to within 1 week for 62.8% and 53.4% of growth-restricted infants, respectively.

## Performance of gestational age estimation models using cord blood data

As with heel prick data, algorithmic estimates of gestational age most accurate among term infants (*Table 2*). When applied to cord blood-derived data, the baseline model (Model 1) and model including analytes (Model 2) performed comparably (RMSE of 1.51 weeks and 1.45 weeks, respectively). As with heel prick data, the full model (Model 3) provided the best estimates of gestational age (RMSE of 1.23). Here, gestational age was correctly estimated to within 2 weeks for 90.4% of infants overall (90.7% and 85% for growth-restricted infants with birthweight below the 10th and 3rd percentiles, respectively; 84.2% for infants < 2500 g). A comparison of the two sample types

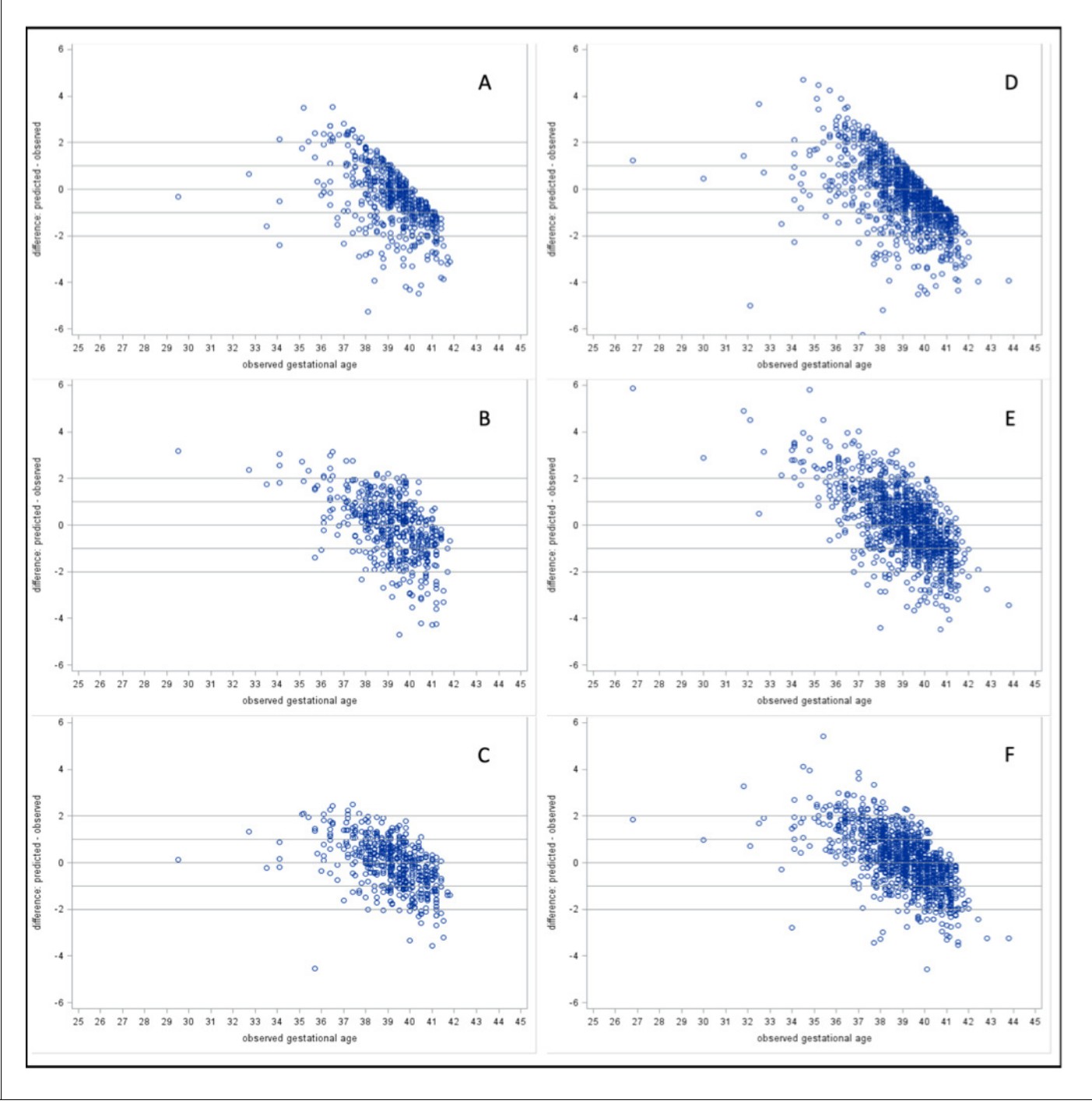

**Figure 2.** Residual plots of predicted – observed by observed gestational age. Heel prick samples: (**A**) Model 1: Baseline Model, (**B**) Model 2: Analyte Model, and (**C**) Model 3: Full Model. Cord blood samples: (**D**) Model 1: Baseline Model, (**E**) Model 2: Analyte Model, and (**F**) Model 3: Full Model.

DOI: https://doi.org/10.7554/eLife.42627.005

indicated that metabolic dating models using data derived from heel prick samples provided more accurate gestational age estimates than models using cord blood samples.

**Table 2.** Proportion of samples with gestational age correctly estimated within 1 week, 2 weeks of ultrasound-validated gestational age.

| | | Heel prick samples | | | | Cord blood samples | | | |
|---|---|---|---|---|---|---|---|---|---|
| | | Overall, n(%) | SGA10, n(%) | SGA3, n(%) | <2500 g, n(%) | Overall, n(%) | SGA10, n(%) | SGA3, n(%) | <2500 g, n(%) |
| Model 1: Baseline Model | RMSE | 1.46 | 1.76 | 2.32 | 2.22 | 1.51 | 1.82 | 2.38 | 2.21 |
| | n(%) within 1 week | 267 (54.8) | 103 (44.6) | 17 (14.4) | 25 (28.4) | 549 (53.0) | 180 (42.5) | 31 (14.4) | 61 (37.0) |
| | n(%) within 2 weeks | 408 (83.8) | 177 (76.6) | 64 (54.2) | 54 (61.4) | 861 (83.1) | 318 (75.0) | 111 (51.6) | 112 (67.9) |
| Model 2: Analyte Model | RMSE | 1.35 | 1.40 | 1.38 | 1.47 | 1.45 | 1.43 | 1.48 | 1.94 |
| | n(%) within 1 week | 279 (57.3) | 123 (53.4) | 64 (54.6) | 38 (43.2) | 544 (52.5) | 221 (52.0) | 113 (52.5) | 62 (37.6) |
| | n(%) within 2 weeks | 431 (88.5) | 204 (88.1) | 104 (88.1) | 74 (84.1) | 874 (84.4) | 362 (85.4) | 181 (84.1) | 116 (70.3) |
| Model 3: Full Model | RMSE | 1.07 | 1.12 | 1.30 | 1.21 | 1.23 | 1.20 | 1.40 | 1.44 |
| | n(%) within 1 week | 311 (63.9) | 145 (62.8) | 63 (53.4) | 52 (59.1) | 615 (59.4) | 267 (63.1) | 116 (54.1) | 88 (53.3) |
| | n(%) within 2 weeks | 459 (94.3) | 218 (94.3) | 108 (91.4) | 83 (94.3) | 937 (90.4) | 385 (90.7) | 183 (85.0) | 139 (84.2) |

Data are presented as the percentage of the number correctly classified within the total of each birthweight category. Counts were based on the average from 10 imputations rounded to the closest integer.

DOI: https://doi.org/10.7554/eLife.42627.006

## Dichotomous discrimination of gestational age

We evaluated the discrimination of gestational age across a dichotomous preterm birth threshold ($\geq$37 weeks vs <37 weeks gestational age) (*Figure 3*). Gestational age estimation models performed best when applied to metabolic profiles derived from heel prick samples. For both types of samples, the best performance was achieved by the full model containing all clinical and analyte data (Model 3) (area under the curve [AUC] 0.945 (95% CI 0.890, 0.999) for heel prick profiles and AUC 0.894 (95% CI 0.853, 0.935) for cord blood profiles).

## Discussion

In this paper, we demonstrate that algorithms developed using newborn screening data from Ontario, Canada are effective in deriving estimates of gestational age in infants born in Matlab, Bangladesh that are accurate to within approximately 1 to 2 weeks of ultrasound-validated gestational age. Data derived from newborn heel prick samples consistently yielded more accurate estimates of gestational age than cord blood-derived data, likely reflecting the fact that our models were originally developed from data obtained from this sample type. Indeed, we have shown that the correlation between cord blood and heel prick-derived data varies significantly across analyte subtypes (Appendix 1).

Accurate assessment of gestational age, preterm birth and small for gestational age is a recognized priority area where there is a need to improve program tracking and accountability (*March of Dimes, 2012*; *WHO, 2014*). Although birthweight data are collected in most settings, it is an unreliable surrogate for gestational age that is prone to overestimation of preterm birth rates in low- and middle-income settings where a high proportion of infants are born small for gestational age. Commonly-used gestational age assessments applied after birth are hampered by their reliance on complex scoring systems. A recent systematic review and meta-analysis of 18 newborn assessments

**Table 3.**

Areas under the ROC curve (AUC) for Bangladesh heel prick and cord blood models, and Ontario reference models.

| | AUC (lower, upper 95% confidence limits), | | |
|---|---|---|---|
| | A) Model 1: Sex, Multiple Birth Status, Birthweight Model | B) Model 2: Analytes, Sex, Multiple Birth Status Model | C) Model 3: Full Model |
| | 0.840 (0.754, 0.925) | 0.895 (0.823, 0.968) | 0.945 (0.890, 0.999) |
| Bangladesh Cord | 0.806 (0.755, 0.858) | 0.823 (0.773, 0.873) | 0.894 (0.853, 0.935) |
| Ontario Reference (*Wilson et al., 2017*) | 0.915 (0.909, 0.921) | 0.946 (0.941, 0.952) | 0.967 (0.963, 0.971) |

DOI: https://doi.org/10.7554/eLife.42627.008

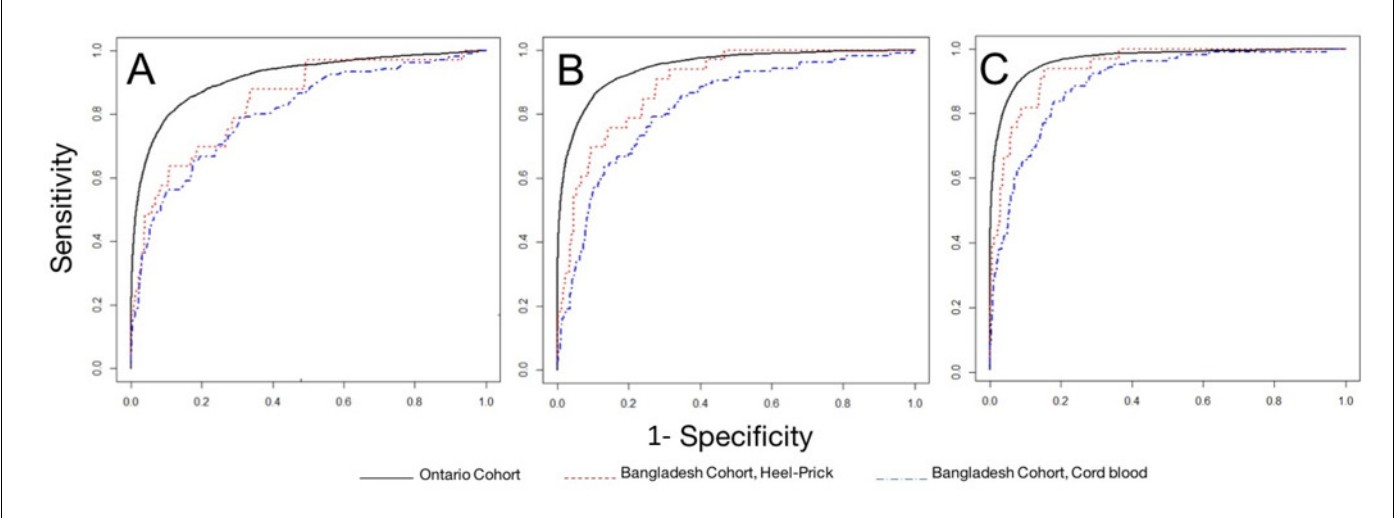

**Figure 3.** Performance of models to correctly classify infants according to dichotomous preterm birth threshold (37 weeks gestational age). Receiver operator curves for: (**A**) Model 1: Heel prick AUC 0.840 (95% CI 0.754, 0.925), Cord blood AUC 0.806 (95% CI 0.755, 0.858); (**B**) Model 2: Heel prick AUC 0.895 (95% CI 0.823, 0.968), Cord Blood AUC 0.823 (95% CI 0.773, 0.873). (**C**) Model 4, Heel prick AUC 0.945 (95% CI 0.890, 0.999), Cord Blood AUC 0.894 (95% CI 0.853, 0.935). Receiver operator curves for models applied to a cross-section of Ontario-derived heel prick samples (*Wilson et al., 2017*) are provided for comparison.

DOI: https://doi.org/10.7554/eLife.42627.007

based on a variety of neuromuscular, physical and other criteria determined that the most popular scoring systems (the Ballard and Dubowitz scores) systematically overestimated gestational age with wide margins of error (*Lee et al., 2016*). Whereas gold standard first trimester ultrasound scans are accurate to within one week, the accuracy of measurements based on newborn examination varies from 2 to 4 weeks. Furthermore, newborn clinical assessments of gestational age such as Dubowitz and Ballard scoring, and neonatal anthropometrics have been demonstrated to be inaccurate surrogate markers of gestational age, specifically in rural communities of Bangladesh (*Lee et al., 2016*).

Metabolic gestational dating approaches emerged in response to the urgent need to improve the epidemiology and surveillance of preterm birth. Circulating newborn metabolites are known to be affected by gestational age and gestational age is routinely considered in the interpretation of newborn screening analysis (*Slaughter et al., 2010*; *Oladipo et al., 2011*; *Newborn Screening Ontario, 2017*). To date, three groups in North America have developed metabolic dating algorithms based on newborn health administrative datasets (*Jelliffe-Pawlowski et al., 2016*; *Ryckman et al., 2016*; *Wilson et al., 2016*). Research has since sought to refine existing models through the addition of analytes known to correlate with gestational age and develop tiered models of varying complexity. Our own group has demonstrated that proportions of fetal and adult hemoglobins are some of the strongest individual predictors of gestational age, (*Wilson et al., 2017*) and we have also validated our algorithms across ethnic subgroups in Ontario (*Hawken et al., 2017*). Efforts are currently underway to begin implementing metabolic gestational age dating in low-resource settings to determine the burden of preterm birth and intrauterine growth restriction. The results from our study offer a reason to be optimistic about these efforts. While the intent of metabolic gestational age dating at present is to provide population-based estimates of the burden of preterm birth, it is conceivable that this approach could also be used to guide care for individual newborns who are identified as preterm.

Our study had a number of important strengths and limitations. Strengths of our approach include the use of internationally-derived samples to externally validate our models and using samples from a well-described cohort of infants with gestational age confirmed by first trimester ultrasound. The study design of the PreSSMat cohort in which our study was nested ensured that enrollment was open to a representative selection of women and newborns delivering in the Matlab icddr,b service area. Other strengths include the high quality of samples received for analysis, and the use of paired cord blood and heel prick samples to compare model performance metrics. The

primary limitation of this study is the participation bias against very preterm and extremely preterm infants, whose parents expressed reluctance to subjecting their newborn to these collection procedures. As a result, we had a relatively small number of samples collected from very preterm and extremely preterm infants, limiting our ability to comment on model performance in these subgroups. In this Bangladesh cohort, the gestational ages estimated from our models were most accurate in infants who were confirmed to be close to full-term by first trimester ultrasound. Algorithm-derived gestational ages tended to be overestimated in preterm infants and underestimated in post-term infants. This suggests that calibration in the large (i.e. introducing a calibration slope adjustment (*Steyerberg, 2010*) to model predictions could improve overall model performance in this external cohort, although this was not conducted in the current study.

Our findings are encouraging for several reasons. First, this work provides early evidence that gestational dating models developed using metabolic data derived from a North American cohort perform well in low-resource populations. The model originally published by our group was developed using data from a Canadian-born cohort of 250,000 infants. In Ontario, the model was able to estimate gestational age to within one week (RMSE 1.06 vs 1.07 for the Bangladeshi cohort) overall and correctly ascertain gestational age to within 2 weeks for 94.9% of infants (vs. 94.3% for the Bangladeshi cohort) (*Wilson et al., 2016*); estimates that compare favorably against other currently-used postnatal gestational age estimation methods that produce estimates varying in accuracy from 2 to 4 weeks gestational age (*Taylor et al., 2010*; *Spinnato et al., 1984*; *Robillard et al., 1992*; *Lee et al., 2016*; *Alexander et al., 1992*). Second, our metabolic models provided significantly improved estimates of gestational age among infants with birthweights < 2500 g, cases where current estimates based on symphysis fundal height and neuromuscular assessments perform poorly (*Spinnato et al., 1984*; *Goto, 2013*). Lastly, we are encouraged by the potential utility of cord blood profiles for deriving gestational age estimates. Differences in cord blood and heel prick profiles described in our analysis likely stem from a number of factors related to timing of collection, including early postnatal fluctuations in neonatal TSH levels, (*Ryckman et al., 2012*; *Büyükgebiz, 2013*) and infant feeding status prior to collection. Although the performance of the models when applied to cord-blood-derived data was somewhat attenuated relative to heel prick data, development of cord-blood-specific models restricted to analytes less susceptible to fluctuations in the postnatal environment may further improve gestational age estimation.

Ultimately, acceptable levels of error in gestational age measurements will need to be determined by public health and maternal child health officials. Given the acknowledged limitations of existing alternatives to ultrasound estimation, metabolic gestational dating approaches appear to offer reliable estimates that are unencumbered by user variability. As we prepare for the scale-up and implementation of metabolic gestational dating approaches for robust population-level estimates of preterm birth, our findings highlight a number of opportunities and challenges. First, heel prick samples taken for newborn screening are typically collected at least 24 hr after birth to accommodate postpartum fluctuations in analyte levels. In many settings around the world, mother-infant pairs are discharged from healthcare settings within the first 24 hr after delivery (*Campbell et al., 2016*). As a result, the accuracy of existing metabolic dating algorithms would be compromised by the change in timing of sample collection. Second, newborn screening is not a standard service of practice in low- and middle-income countries, including Bangladesh. It was therefore unsurprising that anecdotal feedback from field nurses assisting with this study indicated that parents were hesitant to consent to heel prick procedures for their infants. Although on-site research staff received extensive training through videos, visual guides and in-person training, a preference for collection of cord blood samples over heel prick amongst research staff may also have affected the number and quality of samples collected. A quality assurance trial was required to improve sample collection and handling techniques. While our current models were originally optimized for application to heel prick data, we highlight an opportunity to optimize these algorithms for use on cord blood data. Transitioning to cord blood-based models would additionally bypass the need to impose discomfort on the child, stress on parents and staff, and also avoid the requirement for extensive training and screening of sample collection techniques. Finally, population-level metabolic screening provides the additional opportunity to provide insight into the prevalence of congenital conditions in participating jurisdictions.

In summary, metabolic gestational age dating approaches offer a novel means for providing accurate population-level gestational age estimates. As we work toward implementing preterm birth

surveillance initiatives in a variety of low-income settings (*Mundel, 2017*), the level of acceptable accuracy of metabolic algorithms should be considered. Application of models to cord blood metabolic profiles is the most feasible option at present, although derivation and optimization of such models are warranted. Utility of other maternal, pregnancy and infant factors that were not available to us in the current analysis for improving existing metabolic dating models may also be of benefit. Where population-level surveillance of preterm birth might be supported through the analysis of a few drops of blood taken shortly after birth, future work should aim to derive models that determine other priority birth outcomes.

## Materials and methods

### Objectives

Our objective was to validate the performance of previously published gestational age estimation models developed in Ontario, Canada (*Wilson et al., 2016*; *Wilson et al., 2017*) in a cohort of infants born in Bangladesh. Specifically, we sought to compare estimates of gestational age derived from our algorithms, through the analysis of newborn blood spots, against estimates of gestational age determined by first-trimester ultrasound. A version of the protocol for this study has been published (*Murphy et al., 2017*). Due to logistical challenges in initiating the study, fewer samples were collected than initially anticipated in our protocol and low numbers of infants with gestational age below 34 weeks. Our methods of sample collection and analysis remained the same.

### Newborn screening

Newborn screening is a public health initiative that screens for rare, treatable conditions that typically produce no symptoms in the neonatal period. Programs vary in scope by jurisdiction, screening for one to over 50 conditions (*Therrell et al., 2015*). In Ontario, as in many regions, drops of blood are taken by infant heel prick, typically within the first few days after birth, and dried onto filter paper. Dried blood spot samples are then analyzed by a series of assays including tandem mass spectrometry, colorimetric and immunoassays as well as high-performance liquid chromatography for metabolic, genetic and other analyte markers.

### Study design

Sample collection was conducted in the Matlab sub-district of Chandpur, Bangladesh where the International Centre for Diarrhoeal Disease Research, Bangladesh (icddr,b) has been running a Health and Demographic Surveillance System (HDSS) in Matlab since 1966. Based on service provision, the HDSS area is divided into two jurisdictions: 1) the icddr,b service area where women of reproductive age and their children under 5 years of age receive care though icddr,b facilities; and 2) the government service area where individuals receive care from government facilities as in other areas of the country. The present study was conducted in the icddr,b service area, and nested within a cohort study entitled 'Preterm and Stillbirth Study, Matlab' (PreSSMat) that was designed to capture data on the biological determinants of adverse pregnancy outcomes, including preterm births. In the PreSSMat cohort, pregnant women were followed prospectively along the pregnancy continuum, with scheduled visits at 11–14 (enrollment and ultrasound), 22–24, and 32 weeks' gestation, at delivery, and at 6 weeks post-partum to collect socio-demographic and clinical data as well as biological specimens. Preterm births were defined as all births that occurred at <37 weeks' gestation. 'Very preterm births' were those that occurred at <32 weeks, and 'extremely preterm births' were those that occurred at <28 weeks. Small for gestational age (SGA10) was defined as cases where birthweight was below the 10th percentile within categories of week of gestational age at delivery and infant sex. The percentiles were calculated and applied based on a North American distribution of birthweight within sex and gestational age categories. We also calculated SGA3, which identifies infants below the 3rd percentile within gestational age and sex categories and is much more likely to reflect infants who suffered intrauterine growth restriction, especially in low and middle-income countries such as Bangladesh where birthweights are lower. Pregnant women were identified by community health workers through monthly home visits. All enrolled women underwent a gestational dating ultrasound at enrollment; otherwise no explicit inclusion or exclusion criteria were applied. All women enrolled in the PreSSMat cohort were eligible for participation in the current study.

## Sample collection and analysis

To examine the effect of timing of sample collection on newborn metabolic profiles, cord blood was collected immediately after birth and spotted on Whatman 903 filter paper. A second dried blood spot sample was also collected via heel prick within 72 hr of delivery or immediately prior to discharge, whichever happened first. The latter reflects the timing of collection for samples used to develop our previously published gestational age estimation models (recommended timing of sample collection for healthy newborns in Ontario, Canada is 24–48 hr after birth). Samples were collected onto filter paper, air-dried and shipped weekly to Newborn Screening Ontario (NSO), the provincial newborn screening facility in Ottawa, Canada. Samples were stored in a temperature and humidity-controlled environment prior to shipment. Eight 3.2 mm diameter samples were punched from each sample for testing of the following analytes: hemoglobin profiles; 17α hydroxyprogesterone (17-OHP); thyroid stimulating hormone (TSH); immunoreactive trypsinogen (IRT); a panel of 12 amino acids and 31 acylcarnitines; t-cell receptor excision circles (TREC); biotinidase activity; and galactose-1-phosphate uridylyltransferase activity. Hemoglobin profiles were determined by high-performance liquid chromatography on a Bio Rad Variant nbs system; neonatal 17-OHP, TSH and IRT were measured using PerkinElmer AutoDELFIA Immunoassays; amino acid and acylcarnitine analysis was performed by electrospray ionization tandem mass spectrometry (Waters TQD); total TREC copy number was measured by quantitative polymerase chain reaction using a ThermoFisher Scientific Viia 7; biotinidase and galactose-1-phosphate uridyltransferase levels were measured using the Astoria-Pacific SPOTCHECK Pro system. Clinical covariates were retrieved from the PreSSMat database to facilitate clinical interpretation of newborn screening data, and also for inclusion as model parameters in this study. *Figure 4* summarizes the study design.

Newborn screening blood spots are subject to degradation if collected or handled inappropriately. Samples with insufficient good-quality dried blood to complete the full panel of assays were excluded from analysis. Samples with missing analyte values had the missing levels imputed (see Appendix 1 for details). In the process of applying newborn screening procedures for the analysis of samples, results of 'screen negative' and 'screen positive' were generated for conditions screened for by the NSO program. Management of incidental clinical findings (screen-positive cases) has been reported elsewhere (*Murphy et al., 2017*).

## Statistical analyses

### Validation of algorithms

We sought to compare estimates of gestational age and preterm birth based on our analysis of blood spots against first trimester ultrasound estimates, which are considered the gold standard for gestational age measurement (*Committee on Obstetric Practice, the American Institute of Ultrasound in Medicine, and the Society for Maternal-Fetal Medicine, 2017*). The performance of the following models was assessed:

- Model 1: Baseline model containing only the clinical factors of infant sex, birthweight, and multiple birth (yes/no)
- Model 2: Analytes model including infant sex, multiple birth (yes/no) and newborn screening analytes and pairwise interactions including acylcarnitines, amino acids, endocrine and enzyme markers.
- Model 3: Full model containing both clinical and analyte data (infant sex, multiple birth (yes/no), birthweight, newborn screening analytes and pairwise interactions)

Statistical modeling approaches are described in Appendix 1. In brief, sample data were scored using multivariable models previously developed using heel prick blood spot samples in a large cohort of infants born in Ontario, Canada (*Wilson et al., 2017*). The fitted models used for scoring the Bangladesh data included numerous main effects and interaction terms including both analytes and clinical measures (sex, multiple birth, birthweight). However, there was a subset of predictors that were clearly the strongest contributors to the model in terms of independent contribution to explained variance: birthweight (in base and full models) and fetal/adult hemoglobin ratio, TSH, 17OHP, ALA, c5, C4DC and TYR (in the sex +multiple birth +analytes and full models). All analyses were conducted using SAS 9.4 (*SAS Institute, 2017*) and R 3.3.2 (*R core team, 2017*).

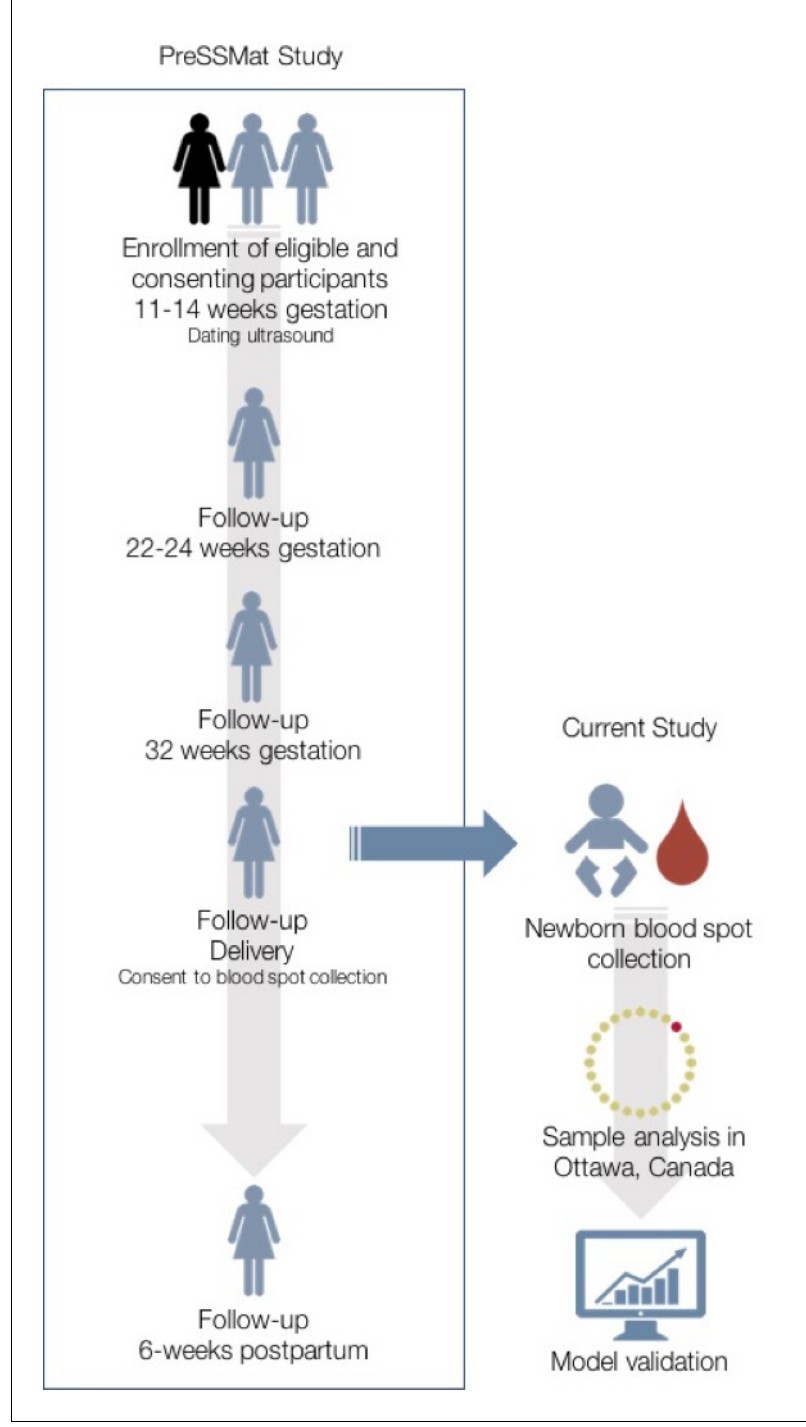

**Figure 4.** Overview of study design. The current study was nested within the PreSSMat cohort operating in Matlab, Bangladesh. Samples were collected from infants born into the cohort and sent to Ottawa, Canada for analysis at a provincial newborn screening facility.
DOI: https://doi.org/10.7554/eLife.42627.009

## Informed consent and ethical approval

Mothers provided informed consent for their infants to be included in the PreSSMat birth cohort and to have clinical data, cord blood and newborn heel prick samples collected and analysed. The present study was approved by the Research Review and Ethical Review Committees of the International

Centre for Diarrhoeal Disease Research, Bangladesh (PR-16039) on July 10, 2016. Approvals were also obtained from the Research Ethics Boards of the Ottawa Health Science Network (20160219–01H) on June 10, 2016, and the Children's Hospital of Eastern Ontario (16/20E) on June 8, 2016.

## Acknowledgements

The authors acknowledge the analytical, lab and other support personnel at Newborn Screening Ontario and the research and clinical staff at icddr,b for their contributions to this project. The funder of the study had no role in the study design, data collection, data analysis, data interpretation, or writing of the report. The corresponding author had full access to all the data in the study and had final responsibility for the decision to submit for publication. JL holds a tier 1 Canada Research Chair in Human Genome Epidemiology.

## Additional information

### Funding

| Funder | Grant reference number | Author |
| --- | --- | --- |
| Bill and Melinda Gates Foundation | OPP1141535 | Kumanan Wilson |

The funders had no role in study design, data collection and interpretation, or the decision to submit the work for publication.

### Author contributions

Malia SQ Murphy, Writing—original draft, Project administration, Writing—review and editing; Steven Hawken, Conceptualization, Data curation, Formal analysis, Validation, Methodology, Writing—review and editing; Wei Cheng, Data curation, Formal analysis, Visualization, Methodology, Writing—review and editing; Lindsay A Wilson, Monica Lamoureux, Courtney Gravett, Project administration, Writing—review and editing; Matthew Henderson, Formal analysis, Writing—review and editing; Jesmin Pervin, Anisur Rahman, Investigation, Writing—review and editing; Azad Chowdhury, Eve Lackritz, Beth K Potter, Mark Walker, Julian Little, Writing—review and editing; Pranesh Chakraborty, Conceptualization, Investigation, Writing—review and editing; Kumanan Wilson, Conceptualization, Supervision, Funding acquisition, Writing—review and editing

### Author ORCIDs

Malia SQ Murphy (iD) http://orcid.org/0000-0002-4566-4957
Steven Hawken (iD) http://orcid.org/0000-0002-3341-9022
Wei Cheng (iD) http://orcid.org/0000-0002-1475-4079
Lindsay A Wilson (iD) http://orcid.org/0000-0002-9910-3338
Kumanan Wilson (iD) http://orcid.org/0000-0002-1741-7705

### Ethics

Human subjects: Mothers provided informed consent for their infants to be included in the PreSSMat birth cohort and to have clinical data, cord blood and newborn heel prick samples collected and analysed. The study was approved by the Research Review and Ethical Review Committees of the International Centre for Diarrhoeal Disease Research, Bangladesh (PR-16039) on July 10, 2016. Approvals were also obtained from the Research Ethics Boards of the Ottawa Health Science Network (20160219-01H) on June 10, 2016, and the Children's Hospital of Eastern Ontario (16/20E) on June 8, 2016.

### Decision letter and Author response

Decision letter https://doi.org/10.7554/eLife.42627.013
Author response https://doi.org/10.7554/eLife.42627.014

## Additional files

### Supplementary files

• Source data 1. Numerical data files for Figures 1 and 2.
DOI:

• Transparent reporting form
DOI: https://doi.org/10.7554/eLife.42627.010

### Data availability

Data cannot be made publicly available according to our institutional guidelines on human subject research, but anonymized data and materials can be made available upon request to Dr Wilson and Dr Hawken. These requests will not need to be reviewed by our ethics board, but should be provided in writing. Numerical data files for Figures 1 and 2 are included as Source data 1.

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

## Appendix 1

DOI: https://doi.org/10.7554/eLife.42627.011

## Statistical modelling

### Preparation of validation data

For samples with missing analyte values, values were imputed by multiple imputation via predictive mean matching (**van Buuren, 2013**; **Harrell, 2001**). This approach borrows information from other study participants with similar patterns for non-missing analytes, using a regression approach to impute missing values using observed values from the other participants. Gestational age (the outcome) and birthweight (the strongest correlate with outcome) were not included as predictors in the predictive mean matching process, to avoid all real or perceived risk of inducing improved predictive performance in the external validation data (**Moons et al., 2006**). Ten imputations were done to evaluate the variability across imputations, and the range of calculated model validation performance across the ten imputations was calculated.

In preparation for modeling, newborn screening analytes were winsorized using an adapted 'Tukey Fence' approach (**Tukey, 1977**). In brief, the interquartile range for each analyte was calculated, and analyte levels in excess of four interquartile ranges above the third quartile (the upper Tukey fence) or four interquartile ranges below the first quartile (the lower Tukey fence), were assigned either the Tukey fence value, or the smallest/largest observed value in the dataset, whichever was the least extreme. By this approach we preserved the 'extremeness' of extreme outliers but reduced the influence of values so extreme that they might disproportionately influence model building and parameter estimation.

Finally, analyte levels and newborn birthweights were standardized to have mean of 0 and standard deviation of 1 by subtracting the mean for each analyte and dividing by the standard deviation in each cohort. This had the effect of normalizing analyte and birthweight results for local factors, while preserving relative covariation of analytes and birthweights across the spectrum of observed gestational age.

### Scoring of external validation data and evaluation of model performance

Bangladesh validation data were prepared similarly to Ontario training data and gestational age estimation models previously fit in the Ontario training data were used to score the validation data. Model parameters estimated in the Ontario training data were fixed, and the model regression equation used to calculate an estimated gestational age in the Bangladesh validation data. Model performance was externally validated by comparing the estimated gestational age to the actual ultrasound-validated gestational age of each participating infant. For evaluation of continuous gestational age estimation, root mean square error (measured in weeks) was calculated, as was the percentage of infants with gestational ages correctly estimated within 7, 10 and 14 days of ultrasound-validated gestational age. Performance characteristics for estimating gestational age across dichotomous gestational age thresholds ($\geq$37 vs <37 weeks gestational age) were evaluated using area under the receiver operator curve from a binary logistic regression model.

