## [Decision Letter]

Thank you for submitting your article "Postnatal gestational age estimation using newborn metabolic profiles in Matlab, Bangladesh" for consideration by *eLife*. Your article has been reviewed by three peer reviewers, and the evaluation has been overseen by a Reviewing Editor and Prabhat Jha as the Senior Editor. The following individuals involved in review of your submission have agreed to reveal their identity: Jessica Duby (Reviewer #1); Eric Ohuma (Reviewer #2).

The reviewers have discussed the reviews with one another and the Reviewing Editor has drafted this decision to help you prepare a revised submission.

Summary:

In this manuscript by Murphy et al., the authors study the accuracy of a metabolic based model to determine gestational age for neonates in Bangladesh when compared to a first trimester ultrasound. The authors make a strong case for the potential impact of a successful algorithm in determining population rates of preterm birth and small-for-gestational age in low-resource settings. This is a potentially important study but the description of the study design and execution is far from complete.

In addition, whilst there is no doubt an accurate estimate of GA is important, there are fundamental problems with this approach that is based on postnatal estimation of GA. First, it helps to perpetuate a standard of care that even WHO has abandoned, namely managing pregnancies without knowing the due date. This is the reason the new WHO guidelines recommend at least one ultrasound scan, in part to estimate gestational age prenatally.

In addition, the authors do not properly address the model's ability to distinguish small-for-gestational age (SGA) neonates, which should be one of the model's primary advantages over using birth weight alone as a surrogate marker for gestational age.

Essential revisions:

1) There are deviations from the cited, published protocol that remain unexplained by the authors. (Murphy et al., 2017)

Major differences between published protocol and current manuscript:

a) Sample size: The protocol's sample size requirements are calculated to be 3,500 participants. The current manuscript details 1523 samples from 1069 newborns with no discussion regarding the achieved sample size. As part of addressing sample size differences, it may benefit the authors to present a figure detailing number of eligible participants, number approached for consent, number consented, etc.

b) Models: The proposed models to test in the protocol are quite different than the models tested in the current manuscript and no justification is provided. For reference, the models listed in the protocol are: "1) birth weight alone; 2) combination of birth weight and fetal/adult haemoglobin levels; 3) combination of birth weight, haemoglobin levels, thyroid-stimulating hormone and 17-OHP (all non-mass-spectrometry-derived analytes); and 4) birth weight and the full panel of newborn screening analytes. Sex and multiple birth (yes, no) will be included in all models." (Murphy et al., 2017) For models used in the reported study, refer to the subsection “Validation of algorithms”.

2) SGA/large-for-gestational age (LGA) babies: The authors need to provide further analysis and discussion regarding the algorithm's performance in SGA/LGA identification. Specifically, the authors need to identify how many SGA/LGA neonates were in their sample and the models' accuracy for these important sub-groups. The authors had listed this analysis in their protocol and have previously performed similar sub-group analysis on the original Ontario cohort which can be used as a guide. (Wilson et al., 2016).

3) While the authors do sub-group analyses by birth weight (<2500g and ≥2500g) in Table 2, weight alone is unimportant when we consider the overarching goal of finding a model that can distinguish preterm birth from SGA and post-term birth from LGA. I would recommend replacing the columns under each of the sample types in Table 2 to be:

1) Overall;

2) SGA;

3) Appropriate-for-gestational age (AGA); and

4) LGA.

4) Imputation of missing values: As only 46 (11%) of observations were imputed, did the authors do a sensitivity analysis of the performance of the model with and without the imputed missing values to evaluate their influence on the results?

5) Prediction score (nomogram) – it would be useful for the authors to also show the relative contribution of each factor in their model. For example, in the baseline model with birthweight, infant sex and multiple births, what is the greatest predictor of GA? A quantification of the relative contribution of each of the factors is useful in understanding their contribution and justification as important variables the prediction of GA. Similarly, this should be done for models 2 and 3 as it allows one to make judgements on how much more improvements in prediction is provided for by including the clinical and analyte data. Also, a comparison of scores helps to discern whether some factors included in the model are correlated and thus are similar in their prediction of GA.

6) Would also be helpful for the author to show a plot of true GA vs. predicted GA (calibration) as this will evidently show the variability of the prediction as a function of GA as opposed to the aggregated estimates they have presented by GA in Figure 1.

7) The cohort was apparently nested within a PreSSMat study in Matlab but the authors do not provide key information on timing of ultrasound examination, birth examinations and weight and criteria used to categorize infants as either preterm, small for gestational age (SGA) or combinations thereof. Please add the necessary level of detail to the methodology and Results sections.

8) Given the issues with cord blood specificity and the wide range of times in collection of heel prick samples (from a few hours to up to 40 hours), one is left guessing as to the fidelity of design of the gold standard used in this study and early measures. Please explain when were these measurements obtained?

9) Please explain what standards were used for ultrasound based gestational age and add an appropriate justification for adoption of the chosen standards.

10) The overall rates of prematurity in the cohort determined from the study (both cord and heel prick samples) are implausibly low (given what is known from ANISA and AMANHI studies done in Bangladesh). Please explain the reasons for these differences.

11) The authors should comment on what is known about prematurity incidence from prior studies of maternal antenatal care and nutrition supplementation done in Bangladesh on large rural cohorts including Matlab, without that information, it is difficult to place this study in the context of what else is known.

---

## [Author Response]

Summary:In this manuscript by Murphy et al., the authors study the accuracy of a metabolic based model to determine gestational age for neonates in Bangladesh when compared to a first trimester ultrasound. The authors make a strong case for the potential impact of a successful algorithm in determining population rates of preterm birth and small-for-gestational age in low-resource settings. This is a potentially important study but the description of the study design and execution is far from complete.In addition, whilst there is no doubt an accurate estimate of GA is important, there are fundamental problems with this approach that is based on postnatal estimation of GA. First, it helps to perpetuate a standard of care that even WHO has abandoned, namely managing pregnancies without knowing the due date. This is the reason the new WHO guidelines recommend at least one ultrasound scan, in part to estimate gestational age prenatally.

Thank you for addressing this point. We agree that managing pregnancies without knowledge of gestational age is not recommended. We therefore do not seek to implement our postnatal estimation models as a clinical tool replace prenatal ultrasound. Rather, we propose that our models could have utility in deriving population-level estimates of preterm birth in geographies or locales where prenatal ultrasound data are frequently available. We have noted in the Introduction that these models are intended for use when ultrasound is not available.

“…these algorithms have the potential to provide reliable population estimates of preterm birth burden when prenatal ultrasound is not available”.

In addition, the authors do not properly address the model's ability to distinguish small-for-gestational age (SGA) neonates which should be one of the model's primary advantages over using birth weight alone as a surrogate marker for gestational age.

Thank you. One of the limitations of our best performing model is its reliance on birthweight as a covariate in estimating gestational age. Including birthweight in the models limits our ability to identify SGA or IUGR infants as the impact of birthweight on gestational age is confounded in these infants.

When birthweight is removed from the model, its performance deteriorates substantially among infants overall, but improves slightly in SGA infants. We have included model performance results for infants with birthweight below the 10^th^ and 3^rd^ percentiles for gestational age and sex as defined by a North American population, however a large proportion of these infants are not truly growth restricted in the context of the local Bangladesh population distribution.

Future work will seek to refine our model that excludes birthweight as a covariate in order to improve its utility in these populations. We discuss the utility of our current models with and without birthweight in SGA cases in greater detail below in response to the other SGA-related reviewer questions.

Essential revisions:1) There are deviations from the cited, published protocol that remain unexplained by the authors. (Murphy et al., 2017)Major differences between published protocol and current manuscript:a) Sample size: The protocol's sample size requirements are calculated to be 3,500 participants. The current manuscript details 1523 samples from 1069 newborns with no discussion regarding the achieved sample size. As part of addressing sample size differences, it may benefit the authors to present a figure detailing number of eligible participants, number approached for consent, number consented, etc.

Thank you for giving us the opportunity to address the differences between our published protocol and our completed study. As the reviewer has noted, we ultimately collected and analysed fewer samples than we initially anticipated. This was due to administrative delays at the outset of the project, particularly while the study was reviewed by three research ethics boards in two countries. By the time the study was initiated, there was less time in which to recruit eligible participants before the end of the study. The sample size calculations presented in our protocol were presented across a range of scenarios, and sample sizes of 1500 and 3500 were projections for Zambia and Bangladesh respectively. The projected sample size of 3500 in Bangladesh was based on Power to detect differences in AUCs for models classifying <34 weeks vs. ≥34 weeks GA. We ended up recruiting fewer total subjects and unfortunately almost no infants <34 weeks, so we have only presented results classifying <37 weeks vs. ≥37 weeks. We still have good statistical power to report c-statistics with good precision. For continuous models, sample size estimates were also driven by the desire to have a minimum number of samples to validate using RMSE estimation (e.g., accurate within ± 1 week) in the smallest subgroups of interest. Thus, we cannot report on validation in <34 weeks. For example, where we only had 3 subjects, but we do have sufficient sample in 34-36 week preterm infants as well as term and post-term infants to report our findings among this group. The following statement has been added to the manuscript:

“Due to logistical challenges in initiating the study, fewer samples were collected than initially anticipated in our protocol, leading to low numbers of infants with gestational age below 34 weeks. Our methods of sample collection and analysis remain the same, recognizing these sampling limitations.”

b) Models: The proposed models to test in the protocol are quite different than the models tested in the current manuscript and no justification is provided. For reference, the models listed in the protocol are: "1) birth weight alone; 2) combination of birth weight and fetal/adult haemoglobin levels; 3) combination of birth weight, haemoglobin levels, thyroid-stimulating hormone and 17-OHP (all non-mass-spectrometry-derived analytes); and 4) birth weight and the full panel of newborn screening analytes. Sex and multiple birth (yes, no) will be included in all models." (Murphy et al., 2017) For models used in the reported study, refer to the subsection “Validation of algorithms”.

Models 1, 2 and 3 were conducted as described in the published protocol. This was an editorial omission. All three models included infant sex and multiple birth (yes/no). Model 1 additionally included birthweight, model 2 excluded birthweight and included analytes and pairwise interactions. Model 3 included birthweight, analytes and pairwise interactions in addition to sex and multiple birth (yes/no). This has been corrected in the Statistical Analysis section of the Materials and methods as follows:

“The performance of the following models was assessed:

Model 1: Baseline model containing only the clinical factors of infant sex, birthweight, and multiple birth (yes/no);

Model 2: Analytes model including infant sex, multiple birth (yes/no) and newborn screening analytes and pairwise interactions including acylcarnitines, amino acids, endocrine and enzyme markers;

Model 3: Full model containing both clinical and analyte data (infant sex, multiple birth (yes/no), birthweight, newborn screening analytes and pairwise interactions)”.

The intermediate models that limited analytes to fetal/adult hemoglobin ratio, and fetal/adult hemoglobin ratio+17OHP+TSH were not included in the results because, although these subgroups of analytes were more simply and cheaply measurable, the models including only these analytes did not reach the level of predictive accuracy that would make them useful. For the sake of clarity and parsimony we opted not to report the results for those subsets of analytes, which were initially chosen for pragmatic rather than scientific reasons.

2) SGA/large-for-gestational age (LGA) babies: The authors need to provide further analysis and discussion regarding the algorithm's performance in SGA/LGA identification. Specifically, the authors need to identify how many SGA/LGA neonates were in their sample and the models' accuracy for these important sub-groups. The authors had listed this analysis in their protocol and have previously performed similar sub-group analysis on the original Ontario cohort which can be used as a guide. (Wilson et al., 2016).

Thank you. We address this point and point 3 in one response, below.

3) While the authors do sub-group analyses by birth weight (<2500g and ≥2500g) in Table 2, weight alone is unimportant when we consider the overarching goal of finding a model that can distinguish preterm birth from SGA and post-term birth from LGA. I would recommend replacing the columns under each of the sample types in Table 2 to be:1) Overall;2) SGA;3) Appropriate-for-gestational age (AGA); and4) LGA.

Response to points 2 and 3:

Thank you, we agree that the performance of our models among SGA infants is important. We did not have access to local reference distributions of appropriate weight for gestational age, so we relied on our North American reference distributions, from which we applied cutpoints to define small and large for gestational age (below 3rd percentile, below 10th percentile, and above 90^th^ percentile for given infant sex and GA in weeks. We have added details to the Study Design section of the Materials and methods as follows:

“Small for gestational age was defined as SGA10, for cases where birthweight was below the 10^th^ percentile within categories of week of gestational age at delivery and infant sex. […] We also calculated SGA3, which identifies infants below the 3rd percentile within gestational age and sex categories and is much more likely to reflect infants who suffered intrauterine growth restriction, especially in low and middle income countries such as Bangladesh where birthweights are lower.”

Further, we have added results for SGA10 and SGA3 to Table 2. We have not reported LGA90 since only a very small proportion of infants (3 infants overall) fell into this group.

4) Imputation of missing values: As only 46 (11%) of observations were imputed, did the authors do a sensitivity analysis of the performance of the model with and without the imputed missing values to evaluate their influence on the results?

We did a complete case analysis and investigated the range of results in each of the 10 imputed datasets and determined that the results did not vary substantially in either the complete case analysis, or among the 10 independent imputations. We reported the average performance across the 10 imputed datasets for the validation results.

5) Prediction score (nomogram) – it would be useful for the authors to also show the relative contribution of each factor in their model. For example, in the baseline model with birthweight, infant sex and multiple births, what is the greatest predictor of GA? A quantification of the relative contribution of each of the factors is useful in understanding their contribution and justification as important variables the prediction of GA. Similarly, this should be done for models 2 and 3 as it allows one to make judgements on how much more improvements in prediction is provided for by including the clinical and analyte data. Also, a comparison of scores helps to discern whether some factors included in the model are correlated and thus are similar in their prediction of GA.

Thank you. In the current study, we are using a previously-developed model to score an external cohort for the purposes of external validation and assessment of generalizability. We did not report on the model building and the relative importance of factors in the final model in this study. We did however report on these details in our previously published studies where these models were developed. However, we appreciate the reviewer’s point that this information is important enough to reiterate, so we have added the following to the Statistical Analysis section of the Materials and methods:

“The fitted models used for scoring the Bangladesh data included numerous main effects and interaction terms including both analytes and clinical measures (sex, multiple birth, birthweight). However, there was a subset of predictors that were clearly the strongest contributors to the model in terms of independent contribution to explained variance: birthweight (in base and full models) and fetal/adult hemoglobin ratio, TSH, 17OHP, ALA, c5, C4DC and TYR (in the sex+multiple birtH^+^analytes and full models).”

6) Would also be helpful for the author to show a plot of true GA vs. predicted GA (calibration) as this will evidently show the variability of the prediction as a function of GA as opposed to the aggregated estimates they have presented by GA in Figure 1.

Thank you. In evaluating our models, we generated residual plots across the range of true gestational ages, but did not include these figures in the manuscript. We agree that these plots are very helpful in understanding the performance of these models above and beyond the aggregate RMSE plots we previously included. We have now included a new figure (Figure 2), which is a panel plot of the three models in both heel prick and cord blood models (6 individual residual plots in all).

7) The cohort was apparently nested within a PreSSMat study in Matlab but the authors do not provide key information on timing of ultrasound examination, birth examinations and weight and criteria used to categorize infants as either preterm, small for gestational age (SGA) or combinations thereof. Please add the necessary level of detail to the methodology and Results sections.

Thank you for this opportunity to provide additional details. Within the PreSSMat study, pregnant women visited the local clinic three times during pregnancy. The first visit to obtain an ultrasound occurred between 11-14 weeks’ gestation. Subsequent visits occurred at 22-24 weeks, 32 weeks, delivery, and six-weeks post-partum. Preterm births were defined as all births that occurred at <37 weeks’ gestation. “Very preterm births” were those that occurred at <32 weeks, and “extremely preterm births” were those that occurred at <28 weeks. As noted, above, we did not have access to local reference distributions of appropriate weight for gestational age, so we relied on our North American reference distributions, from which we applied cutpoints to define small and large for gestational age (below 3rd percentile, below 10th percentile, and above 90^th^ percentile for given infant sex and GA in weeks. This has been added to the Materials and methods section:

“In the PreSSMat cohort, pregnant women were followed prospectively along the pregnancy continuum, with scheduled visits at 11-14 (enrollment and ultrasound), 22-24, and 32 weeks’ gestation, at delivery, and at 6-weeks post-partum to collect socio-demographic and clinical data as well as biological specimens. […] The percentiles were calculated and applied based on a North American distribution of birthweight within sex and gestational age categories.”

8) Given the issues with cord blood specificity and the wide range of times in collection of heel prick samples (from a few hours to up to 40 hours), one is left guessing as to the fidelity of design of the gold standard used in this study and early measures. Please explain when were these measurements obtained?

Thank you. As noted above, the gold-standard against which our GA estimates were compared was first trimester ultrasound (see below our response to the following comment for more information). We have noted as a limitation that heel-prick samples should be collected at least 24 hours after birth, but given the rates of early mother-child discharge from hospital, heel-prick samples were often collected before this time out of necessity. The post-partum analyte fluctuations that occur within the first few days of birth are a limitation of cord and early heel-prick sampling, and we have noted that this may have affected the performance of our model. However, as ultrasound-validated GA estimates were obtained between 11-14 weeks’ gestation, we are confident that the gestational ages against which our model was compared were accurate to within one week.

9) Please explain what standards were used for ultrasound based gestational age and add an appropriate justification for adoption of the chosen standards.

Thank you. Ultrasound-based gestational age was determined via first trimester ultrasound (between 11-14 weeks). The American College of Obstetricians and Gynecologists recommends first-trimester ultrasound (up to 14 weeks) as the most accurate method of establishing and measuring gestational age (ACOG, 2017). In this study, a portable ultrasound machine was used for all participants, and included measurement of crown-rump length, biparietal diameter, occipito-frontal diameter, head and abdominal circumferences, and femur length. We have referenced the ACOG recommendations within our manuscript.

10) The overall rates of prematurity in the cohort determined from the study (both cord and heel prick samples) are implausibly low (given what is known from ANISA and AMANHI studies done in Bangladesh). Please explain the reasons for these differences.

Thank you for this comment. We believe there are two primary reasons for the lower rates of preterm birth seen in this cohort. First, the women in this cohort benefited from being in the service area for icddr,b, the organization implementing the study in Matlab. Compared to individuals outside of this service area, women in the icddr,b service area benefit from a high level of prenatal and other health care. Additionally, participation in this study allowed pregnancies to be monitored and antenatal care to be provided in ways that may not have occurred had these women not participated in the study. Second, many parents of very preterm and extremely preterm infants were reluctant to have their infants subjected to the sample collection procedures given the challenges already experienced by these infants, and thus did not consent to participate in the study. This was highlighted previously in our Discussion section, but we have made this limitation more explicit. As our objective was not to determine rates of preterm birth in this area however, we have not expanded further as to why our study contains relatively few preterm infants.

“The primary limitation of this study is the participation bias against very preterm and extremely preterm infants, whose parents expressed reluctance to subjecting their newborn to these collection procedures. As a result, we had a relatively small number of samples collected from very preterm and extremely preterm infants, limiting our ability to validate model performance in these sub-groups.”

11) The authors should comment on what is known about prematurity incidence from prior studies of maternal antenatal care and nutrition supplementation done in Bangladesh on large rural cohorts including Matlab, without that information, it is difficult to place this study in the context of what else is known.

Thank you. We believe that this falls outside the scope of our study, as our aim was not to estimate the incidence of preterm birth in Matlab, but rather to comment on our model’s ability to estimate GA in this context. The next step in the implementation of our model will be to use the model to estimate the incidence of preterm birth, but that was not our goal at this time.